# OpenReview forum: "Breaking MCP with Function Hijacking Attacks: Novel Threats for Function Calling and Agentic Models"
_ICLR.cc/2026/Conference — ICLR 2026 Conference Withdrawn Submission_

### Official Review · Reviewer_FKUb · 2025-10-26

**Soundness:** 3
**Presentation:** 3
**Contribution:** 3
**Rating:** 6
**Confidence:** 4

**Summary:**

The paper introduces FHA against FC language models and MCP-based agentic systems. The core idea is to embed a short adversarial token string inside the natural-language description of an existing tool, so that the model prefers an attacker-chosen function over the ground-truth function when answering a user query. The authors argue these results reveal a previously underexplored threat surface: manipulating tool selection itself rather than only inducing harmful content.

**Strengths:**

- Shifts focus from content jailbreaking to tool-selection manipulation; adversarial payload is confined to tool descriptions, a realistic and under-monitored surface in MCP ecosystems.
- Clear threat model and algorithms; Fig. 1 concretely illustrates the end-to-end impact; Table 2 cleanly compares baselines; appendices provide prompts, algorithms, and additional analyses.
- Highlights an actionable class of attacks for MCP/agentic systems, with immediate implications for guardrail design, tool schema hardening, and deployment practices.

**Weaknesses:**

- The optimized suffixes look like gibberish, which may be caught by basic linting or human review. Quantify a stealth–ASR trade-off by constraining suffixes to syntactically/semantically plausible text and measuring ASR vs. detectability.

- Most experiments use BFCL with ≤4 tools per prompt. Real MCP deployments often expose dozens of tools, OR route through dynamic registries.
- Only open models (2B–7B). Add at least one larger FC-tuned model or a commercial FC API to assess whether the phenomenon persists at scale.
- The universal attack explores transfer across prompt sets, not models. Evaluate training the poisoned description on model A and testing on model B to probe black-box transferability, which matters operationally.

**Questions:**

- Can you constrain the adversarial suffix to dictionary words / natural sentences and report ASR vs. human/automatic detectability?
- Do FHA adversaries transfer across models (train on Llama‑3B, test on Mistral‑7B/Granite‑2B) and across different FC syntaxes?

---

### Official Review · Reviewer_8mnB · 2025-10-29

**Soundness:** 2
**Presentation:** 2
**Contribution:** 3
**Rating:** 4
**Confidence:** 3

**Summary:**

This paper introduces the Function Hijacking Attack (FHA), a novel attack vector for Function Calling (FC) models.
Unlike traditional prompt injection, FHA operates by inserting adversarial tokens into the description of a tool (function). The objective is to manipulate the model into erroneously invoking an attacker-specified function in response to a benign user query. The authors adapt the GCG attack methodology to this new task, demonstrating high Attack Success Rates on the Berkeley Function Calling Leaderboard (BFCL) benchmark against three small LLMs. The authors also introduce FunSecBench, an extension of BFCL, to evaluate robustness.

**Strengths:**

1. **Novel Threat Model**: The core strength of this paper is its proposal of a novel and realistic threat model. Attacking the tool description, a relatively static and potentially overlooked text, instead of the dynamic user prompt is an insightful approach. Table 1 effectively differentiates this work from existing literature.

2. **Clear Problem Formulation**: The paper's motivation is clear. Figure 1 provides an excellent visual explanation of the attack mechanism and its potential harm, allowing readers to quickly grasp the core problem.

3. **Initial Empirical Exploration**: The paper successfully validates the feasibility of FHA on the BFCL dataset. It also provides several useful ablation studies, such as the impact of the adversarial suffix (optim_str) length (Section 6.2) and the function set composition (Section 6.3) on the attack's effectiveness.

**Weaknesses:**

1. **Limited Scope of Experimental Benchmark**: My most significant concern is that the evaluation is conducted only on the BFCL benchmark. As noted in Appendix E (Figure 9), samples in BFCL contain an average of only 2-4 available functions. This setup does not seem representative of the real-world agentic applications, which often involve dozens or even hundreds of MCP tools. Furthermore, the paper's own results (Figure 4B) suggest that adding just one or two 'noise functions' can reduce the attack's efficiency in early epochs. This raises questions about whether the high ASRs are an artifact of the benchmark's simplicity and makes it difficult to assess the attack's viability in more complex, realistic scenarios.

2. **Lack of Evaluation on Relevant Large-Scale Models**: The experiments are confined to three small models (2B, 3B, 7B). The security of function calling and agentic AI is a critical concern, especially for current open-source large models (e.g., Qwen3-14B and Qwen3-32B). While testing on closed-source models may be infeasible, the paper does not demonstrate whether GCG-style attacks remain effective against these larger, more robust open models. Without such an evaluation, the practical significance and generalizability of the findings are limited, as it's unclear if this vulnerability persists beyond small-scale models.

3. **Limited Technical Novelty and Baseline Comparisons**: The FHA method is presented as a direct adaptation of the GCG attack to a new application scenario, with limited algorithmic novelty. The baseline comparisons are also somewhat limited, consisting mainly of 'standard inference' (no attack) and 'function injection' (a baseline implemented by the authors). A direct experimental comparison to other relevant works on adversarial perturbations would strengthen the paper.

4. **Paper Structure and Presentation**: The main body of the paper concludes abruptly after the experiments. Critical analysis, such as the in-depth comparison to related work (e.g., MPMA) and deeper insights on the attack design, is relegated to Appendix A. This content is essential for understanding the paper's contribution and should be integrated into a proper 'Discussion' section in the main text. Additionally, there are minor formatting issues (e.g., inconsistent paragraph indentation on Lines 112, 165, and 168) that suggest a need for further polishing.

**Questions:**

1. **Clarification on Table 2 (Line 303)**: The paper states, "Slot-Filling ASRs for ZS are lower relative to their Function Name ASRs compared to FS". This seems to imply the gap between FN ASR and SF ASR is smaller for ZS than for FS. However, the data in Table 2 appears to show the opposite (e.g., for Granite, the ZS gap is 0.88 - 0.60 = 0.28, while the FS gap is 0.58 - 0.55 = 0.03). Could the authors please clarify this apparent contradiction?

2. **Suggestion Regarding Benchmark Complexity**: My main concern, as stated in the weaknesses, is the simplicity of the BFCL benchmark (2-4 functions). To address this limitation, I suggest performing an evaluation on a more complex setup. Even a preliminary experiment during the rebuttal phase on a sample with, for instance, 30-50 functions would provide much stronger evidence for the attack's real-world viability. A response demonstrating success in this more challenging environment would substantially strengthen the paper.

3. **Suggestion Regarding Model Scale**: The evaluation is currently limited to small models (<= 7B). To demonstrate the broader significance of this vulnerability, I suggest validating the attack on a relevant, larger-scale open-source model (e.g., Qwen3-14B or Qwen3-32B).

4. **Suggestion for Paper Structure**: I recommend moving the critical analysis currently in Appendix A (e.g., the comparison to Zhang et al. and the discussion of MPMA ) into a dedicated 'Discussion' section in the main paper. This content is essential for contextualizing the contribution and should not be relegated to the appendix.

---

### Official Review · Reviewer_wE3i · 2025-10-31

**Soundness:** 3
**Presentation:** 3
**Contribution:** 2
**Rating:** 2
**Confidence:** 3

**Summary:**

This paper presents a new optimization-based adversarial example attack for function hijacking, which forces a target model (LLM) to use an attacker-specified function instead of the ground truth one. Intuitively, this work builds on the optimization-based GCG adversarial attack and repurposes it to select the target function name. The results show that the function hijacking attack (FHA) achieves strong performance with over 90% success rate, outperforming the baseline prompt injection attacks for achieving the same target (which was generated with LLaMA 70b).

**Strengths:**

1. The paper presents strong performance for the FHA in the evaluated settings.
2. Extends a public dataset, namely BFCL, with new variants covering multiple ground truth functions, paraphrasings of the query, and different parameters for the functions.

**Weaknesses:**

1. **Threat Model**: My main concern is the threat model. As I understand, the presented FHA optimizes the adversarial tokens for a specific query and functions order. In practice, I do not think this will happen, as a naive defense could simply shuffle the functions each time. Targeting a very specific query is also a strong assumption.
    - The paper begins to explore adding different functions and the transferability of the attacks towards the end, but I believe this setting should be the default and should be clarified and expanded.
    - There are no experiments involving changing the query, shuffling the function names, and using different arguments, which would be a normal use case unless this is a highly targeted attack.
    - I believe the setting for CGC is a bit different because it's a jailbreak and hence a direct injection attack, where the attacker is the one using the prefix. Here, the attacker is not the one using the prefix but the user. There should be a clear boundary between what the prefix is optimized on and what it is tested on.

2. **White Box Threat Model**: The threat model here assumes a white-box scenario (although I couldn't find this explicit assumption in the threat model section in Section 4, and I would suggest adding it). This is a strong threat model, especially when other prompt injection attacks can be done in a black-box manner.

3. **Weak Baseline for Prompt Injection Attack**:
    - LLaMA 3 70b is used and the attack is performed in a single inference, compared to the optimization-based FHA which can take 500 epochs. Why was LLaMA 70b used instead of a GPT model, which are usually better and were later used in synthetic data generation?
    - Multiple iterations could be performed, providing the model with feedback on whether the attack was successful or not, and giving it the history to achieve better performance for the injection attack.

**Questions:**

1. For the Granite model, I do not understand how the FHA still works when the model itself expects to output the arguments before the function name. Wouldn't this break the parsing tool calling code?  I also don't fully understand the difference between the Arg flip curve and the normal Granite curve in Figure 2. And as minor point: It would be really helpful if the color of one of the curves was made a bit different to distinguish them more easily. Additionally, the font size in the figures is a bit too small; it would be much more helpful if it could be increased.

2. In Figure 4b, the results for 1 noise at the end are better than those for 0 noise. Is there any intuition as to why this happens?

3. In data augmentation strategy 2, does this only change the arguments or also paraphrase? If it only changes the arguments, then why not both?

---

### Official Review · Reviewer_8Ztp · 2025-10-31

**Soundness:** 3
**Presentation:** 2
**Contribution:** 2
**Rating:** 2
**Confidence:** 4

**Summary:**

This paper presents a security threat to agentic AI systems called "function hijacking," where attackers manipulate AI models to invoke specific, malicious functions rather than the intended ones. Unlike previous injection and jailbreaking attacks that target user prompts, this attack exploits the function calling interface that allows AI agents to interact with external tools.
The authors demonstrated the severity of this vulnerability by achieving attack success rates of 80-98% across three different models using the BFCL dataset, and they introduced FunSecBench, a new benchmark for evaluating function calling security.

**Strengths:**

1. This work addresses a really urgent and timely problem. As we're seeing function calling and agentic AI absolutely everywhere right now - just look at how popular MCP and similar frameworks have become. So the fact that the authors are looking at security vulnerabilities in these systems is super important and relevant to what's actually being deployed in the real world today.

2. The experimental work here is thorough. The authors evaluated their attack across three different models and achieved consistently high attack success rates, ranging from 80% to 98%.

**Weaknesses:**

I have a few major concerns about this work:

First, regarding the threat model - Looking at Figure 1, it appears you're assuming the attacker can directly modify the function descriptions themselves. And honestly, given how successful jailbreaking attacks have been, it's not that surprising that you can hijack function calls if you can literally rewrite the function descriptions. The threat model seems quite powerful - maybe too powerful. What I'm really skeptical about is whether this would work in more realistic indirect prompt injection scenarios.

What would be much more interesting and convincing is this: suppose the attacker can only modify one function that they have access to - can they then hijack other tools that they cannot directly manipulate? For example, tools designed by another party, like in the MCP framework. I'm curious whether your method would work in this more realistic, constrained setting.


Section 3  feels totally unnecessary. Especially all those mathematical formulations - they're not really used later in the paper, or at least they don't need to be formalized like that.

Last, about the method - It seems like you're essentially just applying a GCG-like approach, right? But here's what I'm confused about: in GCG, they optimize the model to start its response with something generic like "Sure" - which is universal and transfers well across different prompts. But if you want to call a specific function, say delete_file, that seems very targeted. How do you optimize the suffix tokens in this case? It doesn't seem like it would transfer to making the model call send_email instead. Am I understanding this correctly, or am I missing something about how your optimization works?

**Questions:**

1. Adversarial perturbation in Tab 1. What's the attack goal here? Is it just a jailbreak attack?
2. I guess the GCG attack is kind of old now. Would applying a stronger method in your setting also improve results significantly?

---

### Author Response · Authors · 2025-11-28

Dear reviewers, we thank you for your thoughtful comments, feedbacks and questions. We will take your reviews into consideration to improve our work.

---

### Note · Authors · 2025-11-28

**Comment:**

Dear reviewers, we thank you for your thoughtful comments, feedbacks and questions. We will take your reviews into consideration to improve our work.

**Withdrawal Confirmation:**

I have read and agree with the venue's withdrawal policy on behalf of myself and my co-authors.